# Marginal Bone Loss Compared in Internal and External Implant Connections: Retrospective Clinical Study at 6-Years Follow-Up

**DOI:** 10.3390/biomedicines11041128

**Published:** 2023-04-08

**Authors:** Bianca D’Orto, Carlo Chiavenna, Renato Leone, Martina Longoni, Matteo Nagni, Paolo Capparè

**Affiliations:** 1Dental School, IRCCS San Raffaele Hospital, Vita-Salute San Raffaele University, 20132 Milan, Italy; 2Department of Neurosciences, Reproductive and Odontostomatological Sciences, Division of Fixed Prosthodontics, “Federico II” University of Naples, 80100 Naples, Italy

**Keywords:** dental implants, implant–abutment connection, marginal bone loss, external connection, internal connection

## Abstract

The aim of this study was to assess and compare the marginal bone loss between two different categories of implants (Winsix, Biosafin, Ancona, Italy) having the same diameter and belonging to the Torque Type^®^ (TT^®^) line, in the internal hexagon version (TTi, Group A) and in the external hexagon configuration (TTx, Group B). Patients with one or more straight implants (insertion parallel to the occlusal plane) in the molar and premolar regions in association with tooth extraction at least 4 months prior to implant insertion, who have a fixture diameter of 3.8 mm, who followed up for at least 6 years, and whose radiographic records were available were enrolled in this study. Depending on implant connections (externally or internally), the sample was divided into groups A and B. For externally connected implants (66), the marginal resorption was 1.1 ± 0.17 mm. The subgroup of single and bridge implants showed no statistically significant differences with a marginal bone resorption of 1.07 ± 0.15 mm and 1.1 ± 0.17 mm, respectively. Internally connected implants (69) showed an overall marginal resorption of 0.91 ± 0.17 mm, while the subgroup of single and bridge implants showed resorption of 0.90 ± 0.19 mm and 0.90 ± 0.17 mm, respectively, with no statistically significant differences. According to the obtained results, internally connected implants showed less marginal bone resorption than externally connected implants.

## 1. Introduction

Dental implants can be defined as artificial tooth roots made of biocompatible materials, such as titanium or ceramic, that are surgically placed into the jawbone to support a replacement tooth or bridge. They are used to replace missing teeth and are a popular alternative to dentures or bridges [1].

Fifty years after the studies of Brånemark et al. [2,3], dental implants supporting fixed prostheses can be considered an effective therapeutic solution in the rehabilitation of an edentulous patient, overcoming the limitations of removable prostheses and promoting effective function and esthetics [4].

The concept of osseointegration was introduced by Branemark et al. in the late 1960s. Together with his research group, he succeeded in developing an oral implant whose clinical application was based on a direct anchorage to the bone tissue. The following definition of osseointegration was suggested (1983): “a direct connection, without interposed layers of soft tissue, between the bone and a loaded implant” [2,3].

From a structural point of view, the osseointegration process is characterized by three phases: primary stability, secondary stability, and maintenance of osseointegration under functional loading [5,6]. 

Primary stability is defined as the absence of mobility of the fixture at the osteotomy site immediately after its placement and reflects pure mechanical engagement and is achieved during the first surgical phase. Secondary stability refers to the additional stabilization that occurs after the implant has fully integrated with the surrounding bone tissue. It is primarily achieved through the development of a strong and stable interface between the implant surface and surrounding bone tissue. This interface is known as the implant–bone interface. Secondary stability is essential for the long-term success of dental implants, as it helps to ensure that the implant remains firmly anchored in the jawbone and can withstand the forces of biting and chewing [7].

To switch from primary to secondary stability and thus obtain osseointegration, micromovement levels at the bone–implant interface must be kept below 150 microns [8].

Should these forces exceed the range of micromovements, they are capable of inducing mobility on the implant, even if it is minimal; during healing, this can result in a disturbance in the bone formation process leading to implant failure during the osseointegration phase. Primary stability is, therefore, guaranteed by the bone that surrounds and delimits the implant site and is necessary to avoid micromovements that can lead to the formation of peri-implant fibrous tissue and consequent fibrointegration [8].

To date, implant success depends on the degree and structural integrity of the osseous tissue built up between the implant and surrounding bone, which, in turn, may be influenced by several factors, such as the surface characteristics of the fixture, implant placement procedures, any systemic conditions of the patient, and hygienic maintenance protocols [9,10,11].

In 1981, Albrektsson et al. identified several factors that may influence this bone interface: the surgical procedure technique, patient and surgical site status, implant structural properties, material biocompatibility, and kind of prosthetic load applied [12].

Understanding these factors and appropriately applying them can increase the predictability of osseointegration by minimizing potential implant failure [13].

Before going into detail about the characteristics of implant–prosthetic connections available in clinical practice, here are some concepts regarding the main components of an implant system. 

A dental implant is essentially made up of three parts: an implant or abutment, which is a screw-like structure that is inserted into the bone and can be made of titanium or zirconia; an abutment, which is a small metal or ceramic structure that connects the prosthesis to the implant; and a prosthesis, which is an artificial crown that replaces the natural one [14,15,16].

Implant abutment connection refers to the interface between the dental implant and abutment, which is the component that connects the implant to the prosthetic restoration (such as a crown, bridge, or denture), leading over the years to the development of connections with different geometries to improve the stability of the implant–prosthetic interface [17].

Although there may be several types of connections between the fixture and abutment, i.e., Morse taper, tri-channel, external-hex, and internal-hex connections, the present study focused on internal and external connections. 

The external connection originated with the Branemark system [18]; in this case, the abutment is attached to the implant using a screw that is located on the outside of the implant body. This type of connection is easier to access for maintenance and repairs, but it may be more prone to mechanical complications due to the stress placed on the screw. The external connection provides for it to emerge from the implant platform and the other components to be structured so that they can engage around it and then be fixed by means of the connection screw.

This system has been the subject of studies and improvements but still has limitations: greater prevalence of loosening of the connection screw [19]; greater prevalence of rotational misfit between the implant and abutment [20]; poorer esthetic results [21]; and inadequate microbial sealing [22].

In the internal connection, the abutment is attached to the implant using a screw that is located inside the implant body. This type of connection is less visible and may provide better esthetics, but it may be more difficult to access for maintenance and repairs. This connection can be subdivided into internal hexagon, octagon, or the pure taper defined as Morse taper. This connection was developed to increase the contact area between the abutment and implant to more evenly dissipate forces and increase stability [23].

Several factors may influence the marginal bone loss around dental implants, including patient features (i.e., smoking, lack of hygiene, parafunctional habits, uncontrolled systemic diseases) [24], site-specific (i.e., residual bone quantity and quality, and presence and thickness of keratinized mucosa) [25,26], dental implant characteristics (diameter, surface treatment, and type of connection), and prosthesis design (retention method and number of units). 

Several different dental implants are available with a choice of internal and external connection systems. Both these connections as well as the kind of the implant abutment may influence the marginal bone resorption. The abutment and implant can have equal diameters; alternatively, an abutment with a narrower diameter can be employed (platform-switching concept) [27,28]. As lateral forces are transmitted at the contact point between the implant and abutment, the risk of loosening and screw breakage is increased by generating micromovements that induce bone remodeling [29,30,31]. In addition, the presence of gaps on the implant–abutment surface can cause microleakage and bacterial accumulation, which can compromise the dental implant success [32,33,34,35,36,37]. 

The aim of this retrospective clinical study was to assess and compare marginal bone loss, also known as peri-implant bone loss, which is the loss of bone around the dental implants, between two different implant categories (Winsix, Biosafin, Ancona, Italy) having the same diameter and belonging to the Torque Type^®^ (TT^®^) line, in the internal hexagon version (TTi, Group A) and in the external hexagon configuration (TTx, Group B).

## 2. Materials and Methods

### 2.1. Patients’ Selection

This retrospective study was performed at Dental Clinic, Department of Dentistry, San Raffaele Hospital, Milan, Italy. The ethics committee approval number is 190/INT/2021.

The investigation was conducted according to the principles of the Declaration of Helsinki. STROBE (Strengthening the Reporting of Observational Studies in Epidemiology) guidelines were followed (http://www.strobe-statement.org/ (accessed on 1 April 2019)).

During the period from April 2016 to July 2022, patients with social vulnerability and having implant–prosthetic rehabilitation at the premolar or molar sites were consecutively enrolled.

#### 2.1.1. Inclusion Criteria

The eligibility criteria were as follows: one or more straight implants (parallel insertion to the occlusal plane) in the molar and premolar regions in association with teeth extraction at least 4 months before implant placement; a fixture diameter of 3.8 mm; follow-up of at least 6 years; availability of radiographic records (pre-surgical, post-surgical, and post-prosthetic finalization); any implants previously placed in the implant site; structural integrity of the antagonist arch teeth; absence of any oral and/or systemic diseases; ability to perform home hygiene maintenance; and compliance and adherence to professional hygiene sessions and monitoring checks. 

#### 2.1.2. Exclusion Criteria

The exclusion criteria concerning fixtures were prosthesis combination with existing teeth, post-extraction implants, poor radiographic documentation, proximity to periodontal teeth or undergoing periodontal therapy, and proximity to fixtures affected by peri-implantitis.

Patients with systemic diseases, who were undergoing bisphosphonate therapy or head and neck radiotherapy for less than one year, who were suffering from severe malocclusions or parafunctions, who were unable to comply with home and professional hygiene maintenance protocols, and who were smoking more than 10 cigarettes per day were also excluded.

According to the type of implants placed (external or internal connection), the sample was divided into groups A and B.

### 2.2. Pre-Surgical Protocol

Written informed consent was obtained from all patients prior to the implant–prosthetic rehabilitation study, indicating that the patients were fully aware of the study’s purpose and the procedures involved. The study was also approved by the local ethics committee, which suggests that the study followed ethical guidelines and regulations.

Before surgery, professional oral hygiene was provided to reduce the risk of post-surgical complications such as swelling and pain [38,39]. This indicates that the researchers took the necessary precautions to ensure the safety and well-being of the patients.

All diagnoses were made based on clinical and radiographic examinations. Radiographic examination was performed at two levels—panoramic radiography and cone beam computed tomography (CBCT). These imaging techniques were used to assess the residual bone height and measure fixture size and position.

### 2.3. Surgical Procedure

All surgeries were conducted by the same surgeon with advanced surgical knowledge.

Patients received 2 g amoxicillin and clavulanic acid one hour before and a further 1 g twice daily for one week after surgery (clarithromycin was administered as an option in case of allergy, 2 g before surgery and 1 g twice daily for the following week).

The procedure was conducted under local anesthesia induced by opticaine solution infiltration with adrenaline 1:80,000 (AstraZeneca, Milan, Italy). All procedures were performed according to the same protocol.

The first incision was made along the top of the alveolar ridge, offset to the palatal side to achieve the same level of keratinized mucosa on both sides of the flap. The keratinized mucosa was more than 2 mm in each case, making grafting procedures unnecessary.

Vertical distal and mesial release incisions were, therefore, executed to expose the underlying bone ridge. The full-thickness flap was elevated to preserve the subperiosteal anatomical structures.

A lanceolate drill was employed to drill the cortical bone. A ø 2.00 pilot drill was inserted to provide a positioning location for the implant and define the fixture setting. A locating pin was inserted to verify the implant position, emergence, and angulation. Progressive diameter drills were used up to the diameter of the final fixture. To promote primary mechanical stability, the site was vertically overprepared and transversely underprepared. The implants were placed in the edentulous site 0.5 mm below the bone crest with a minimum insertion torque of 35 Ncm.

The inserted implants belonged to the TT line (Winsix, Ancona, Italy). The TT line is characterized by the same implant body and possibility of having an internal (TTi) or external (TTx) hexagon. At the macromorphological level, these implants are characterized by double-threaded coils and a double principle to facilitate implant insertion with half of the coils. The sulcus at the base of the loop decompresses the bone by dissipating forces and facilitates clot deposition. At the same time, it increases the surface area of the implant facilitating the neoformation of cells. The apex is conical and undersized by 1.3–1.8 compared with the diameter of the implant; it is strongly tapered to obtain an osteotomic effect and facilitate the inclined insertion of the implant even in cases of reduced bone availability. Adaptation of the flap and suturing was carried out with 3-0 non-resorbable sutures (Vicryl; Ethicon, Johnson & Johnson, New Brunswick, NJ, USA).

### 2.4. Post-Surgical Protocol

Immediately after surgery, intraoral X-rays were taken to check the correct positioning of the implant. 

Antibiotic therapy (amoxicillin and clavulanic acid 1 g or clarithromycin 1 g in case of allergy, twice daily for 7 days after surgery) and analgesic treatment (non-steroidal anti-inflammatory drugs, as required) were prescribed for each patient. Mouth rinses with a solution of chlorhexidine-digluconate (0.20%) twice daily for 10 days were recommended [40,41,42]. One week after the surgery, the sutures were removed.

### 2.5. Prosthetic Protocol

Implants were coated for approximately 4 months [43,44]; then a reopening was performed, and the coping screws were substituted with healing screws. Each patient was fitted with a single or multiple acrylic provisional prosthesis (1 or 3 dental units depending on the implant placement). A temporary resin (Fermit, Ivoclar Vivadent, Naturno, Bolzano, Italy) was filled into the screw access holes. 

After a further four months, a single or multiple (1 or 3 dental units depending on the implant placement) metal-ceramic or implant-supported resin final denture consisting of three or four units was fabricated to replace the provisional denture.

Articulating papers (40 μm Bausch) were applied to achieve central contacts on all masticatory units (static occlusion) in the provisional prosthesis and a dynamic, premolar-driven, definitive occlusion [21,45].

### 2.6. Follow-Up

Follow-up visits were conducted 1 week after surgery, at 3 and 6 months, and then once a year for the following 6 years. Each patient was included in a professional oral hygiene program every 4 months to both limit potential complications and monitor and intercept them.

During follow-up visits, intraoral radiographs perpendicular to the long axis of the tooth were taken using the long cone and centring technique (XCP, Dentsplay international, RINN).

The measurements were only taken after image calibration to assess the marginal bone development. The software Digora Optime (Soredex, Tuusula, Finland) was chosen as the analysis platform using the specific measurement tool contained therein. As a first step, the calibration (pixels/mm) of the instrument was performed using the diameter of the survey site plant as the known unit. Next, changes in the peri-implant marginal bone height over time were measured in relation to the most coronal part of the implant fixture and the point of contact between the implant fixture and marginal ridge. To estimate the trend of the bone, a guideline above the shoulder of the implant was assumed to be the reference point for measurement from which a straight line was drawn parallel to the long axis of the implant to the most coronal point at which the bone made contact with the fixture both mesially and distally. These parameters were measured at the time of the implant placement, at 12 months, and once a year thereafter.

### 2.7. Statistical Analysis

Statistical analysis of the collected data was performed with SPSS and Microsoft Excel software (SPSS 22.0, IBM Corp., Armonk, NY, USA and Microsoft Excel, Microsoft Corp., Redmond, WA, USA).

Data were reported as mean ± standard deviation, while comparisons of marginal bone resorption in different periods were conducted with *t*-test (*p* ≤ 0.05 was considered the threshold for statistically significant difference).

The null hypothesis was that there were no statistically significant differences between the groups concerning marginal bone loss.

## 3. Results

Ninety-seven patients (29 women and 68 men) with an average age of 57.7 ± 8.53 years fit the study criteria. A total of 135 implants (Winsix, 69 TTi and 66 TTx) from the same manufacturer (Biosafin, Ancona, Italy) were placed, all with a diameter of 3.8 mm.

A total of 53 implants were placed in the mandible and 82 implants in the maxilla. In the premolar region of the upper jaw, 38 implants were placed, and in the mandible, 32 implants were placed. In the molar region of the upper jaw, 30 implants were placed, and in the mandible, 35 were placed. 

The number of implants placed per sector and the corresponding percentage were reported as follows (Table 1).

Sixty-three patients represent group A (TTi implants) and received a total of 69 implants. Group B (TTx implants) consisted of 57 patients with a total of 66 implants.

For the externally connected implants (66), the global marginal resorption was 1.1± 0.17 mm. 

Twenty implants were placed to support single crowns and 46 to support bridges.

The marginal bone loss of the implants supporting single crowns was 1.07 ± 0.15 mm and that of those supporting bridges was 1.10 ± 0.17 mm.

No statistically significant differences were found between the two categories compared (Table 2).

For the internally connected implants (69), the global marginal resorption was 0.91 ± 0.17 mm.

Thirty implants were placed to support single crowns and 39 to support bridges.

The marginal bone loss of the implants supporting single crowns was 0.90 ± 0.17 mm and that of those supporting bridges was 0.90 ± 0.19 mm.

No statistically significant differences were found between the two categories compared (Table 3).

The two groups were compared after one year (T1) and six years (T2). 

The average bone resorption around the implants over the two-year follow-up was 0.98 ± 0.19 mm of which 91.3% was in the first year.

Marginal resorption was significantly higher around externally connected implants (*p* < 0.001), while there were no statistically significant differences between implants prosthesis alone and as a group (Table 4).

The null hypothesis was rejected.

Of the four interactions predicted and calculated by the test, only one proved to be statistically significant. Specifically, there was a statistically significant interaction effect between the implant type (internal vs. external hexagon) and time (T1 vs. T2) on bone resorption values (*p* = 0.007). This means that the internal hexagon implant has a double advantage. Besides giving lower bone resorption values than the external connection, the internal hexagon implant reacts better to the passage of time than the external hexagon implant.

## 4. Discussion

The inclusion and exclusion criteria of this retrospective clinical study were chosen to focus the results on the role of the chosen implant connections, avoiding additional risk factors influencing peri-implant marginal bone loss such as smoking, lack of hygiene, parafunctional habits, and uncontrolled systemic diseases [8,9,10]. The exclusion criteria concerning fixtures were prosthesis combination with existing teeth, post-extraction implants, poor radiographic documentation, proximity to periodontal teeth or undergoing periodontal therapy, and proximity to fixtures affected by peri-implantitis. Although in the recent systematic review and meta-analysis by La Monaca et al. [46] it is stated that tooth-to-implant-supported prosthesis in partially edentulous patients can be a therapeutic alternative with moderate success, several authors reported various possible complications related to this method compared with traditional prostheses (both tooth-supported and implant-supported), such as a higher implant failure rate, risk of fracture of the prosthetic framework, stress on the dental elements with consequent impairment, and difficulty in hygienic maintenance that can cause mucositis and peri-implantitis, thus affecting marginal bone loss around the fixtures [47,48]. Regarding post-extractive implants, 12 months after functional loading, as reported by several studies, they showed higher marginal bone loss than those placed in the healed bone [49,50]. Implants of which few radiographic documentations were available were excluded: as reported by Cassetta et al. [51] in their prospective cohort study, although intraoral radiographs overestimate marginal bone loss, they might still represent the most significant aid to assess peri-implant bone level changes at different follow-ups. Implants in proximity to periodontal teeth or undergoing periodontal therapy and in proximity to fixtures affected by peri-implantitis were excluded from the study: as reported by several authors, the presence of local risk factors influencing the microbiota can favor the development and progression of peri-implant diseases [52,53]. To limit marginal bone loss as much as possible, in addition to applying the above exclusion criteria, further precautions were taken. Each patient underwent cone beam computed tomography (CBCT) to identify the residual bone height and assess fixture length and location.

As suggested by Chackartchi et al., incorrect implant placement can affect the peri-implant bone level, interfering with the survival rate [54]. The site was vertically overprepared and transversely subprepared to promote primary mechanical stability and the minimum insertion torque of 35 Ncm. As reported by Kotsakis et al. [55], the drilling process according to alveolar bone density and insertion torque can have a significant role in primary implant stability, crestal bone level changes, and bone healing. According to Valles et al. [56], the implants were placed in the edentulous site 0.5 mm below the bone crest as fixtures placed in a subcrestal position can have less marginal bone loss changes when compared with implants equicrestally placed. At the end of the surgical procedure, each patient was placed in a professional oral hygiene program every 4 months to remove plaque, which is considered as a primary etiological factor in the development of peri-implant disease and, consequently, marginal bone loss [57,58,59]. Marginal bone resorption was evaluated by comparing externally connected implants (group A) with internally connected implants (group B). For each group, it was also assessed whether there were differences in marginal bone loss between single crowns and bridges. Overall, the average bone resorption around the implants over the two-year follow-up was 0.98 ± 0.19 mm, of which 91.3% was in the first year. Similar results were reported by Khalaila et al. [60] in their prospective clinical study, in which they evaluated the relationship between Periotest values, marginal bone loss, and stability of single dental implants at 3-year follow-up, reporting an average marginal bone loss at 1 year of 1.732 mm. Concerning partial rehabilitations, confirming what was reported for implant-supported single crowns, Naert et al. [61] identified a higher marginal bone loss during the first year and averaged 0.1 mm for the following years. Gherlone et al. [62], in their retrospective clinical study concerning the rehabilitation of the edentulous posterior maxilla with straight or tilted implants, obtained the same trend of marginal bone loss regardless of the type of the surgical procedure and prosthetic loading protocol. Neither in group A nor in group B was there a statistically significant difference between single crowns and bridges; the same result was obtained when comparing the groups with each other in terms of the type of restoration (single or partial). Marginal resorption was significantly higher around externally connected implants than in internal ones (*p* < 0.001), and the null hypothesis was rejected. Similar results, supported by several authors [34,35], were obtained by Kim et al. [63] in their retrospective clinical study with 4–12 years of follow-up in which they compared the marginal bone loss between external-connection and internal-connection dental implants in posterior areas. A total of 170 patients with 355 implants were included in the study, of which 206 were externally connected and 149 were internally connected. The average marginal bone loss was 0.47 mm and 0.15 mm in the externally and internally connected implants, respectively, recording a statistically significant difference between the groups, thus promoting the application of internal connections whenever possible. Opposing results were obtained by Esposito et al. [64], who evaluated the advantages and disadvantages of implants with the same body and different connection (internal and external). Five years after loading, there were no statistically significant differences in the estimated marginal bone loss between the two groups (difference = 0.14 mm, 95% CI: −0.28 to 0.56, *p* = 0.505), and both groups showed statistically significant bone loss since implant placement: 1.13 mm for externally connected implants and 1.21 mm for internally connected implants. Since the 5-year post-loading data did not show any statistically significant differences between the two connection types, the choice of connection type can be based on clinical preferences.

## 5. Conclusions

Within the limitation of this study, internally connected implants reported less marginal bone resorption when compared with externally connected implants.

According to the obtained results and current literature, internally connected implants might be preferred to reduce peri-implant marginal bone loss. Further clinical trial characterized by a larger sample at higher follow-up could be necessary to confirm the obtained results.

## Figures and Tables

**Table 1 biomedicines-11-01128-t001:** Implant placement sites in relation to implants’ number.

Implant Placement Sites	No. of Implants/% Implants
Maxillary premolar	38 (28.33)
Mandibular premolar	32 (21.67)
Maxillary molar	30 (23.33)
Mandibular molar	35 (26.67)

**Table 2 biomedicines-11-01128-t002:** Category, number of implants, marginal bone loss, and statistical analysis concerning externally connected implants divided according to prosthetic design.

Externally Connected Implants	Category	Number of Implants	Marginal Bone Loss	Statistical Analysis
Prosthetic design	Single crown	20 (30.3%)	1.07 ± 0.15	0.776
	Bridge	46 (69.7%)	1.10 ± 0.17	

**Table 3 biomedicines-11-01128-t003:** Category, number of implants, marginal bone loss, and statistical analysis concerning internally connected implants divided according to prosthetic design.

Internally Connected Implants	Category	Number of Implants	Marginal Bone Loss	Statistical Analysis
Prosthetic design	Single crown	30 (43.4%)	0.90 ± 0.17	0.743
	Bridge	39 (56.6%)	0.90 ± 0.19	

**Table 4 biomedicines-11-01128-t004:** Category, number of implants, marginal bone loss, and statistical analysis according to connection type and prosthetic design.

	Category	Number of Implants	Marginal Bone Loss	Statistical Analysis
Connection type	TTi	69 (51.1%)	0.90 ± 0.18	<0.001
	TTx	66 (48.9%)	1.09 ± 0.17	
Prosthetic design	Single crown	48 (35.6%)	0.87 ± 1.07	0.376
	Bridge	87 (64.4%)	0.94 ± 1.11	

## Data Availability

The data presented in this study are available on reasonable request from the corresponding author.

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
