# Peer review of "Marginal Bone Loss Compared in Internal and External Implant Connections: Retrospective Clinical Study at 6-Years Follow-Up"

_biomedicines, 2023, doi:10.3390/biomedicines11041128_

Round 1

Reviewer 1 Report

TITLE:  Marginal bone loss compared in internal and external implant 2 connections: retrospective clinical study at 6-years follow-up

biomedicines-2167694-

The aim of the present investigation was to evaluate and compare marginal bone loss between two different categories of implants.

GENERAL COMMENTS

The article is in-line with the journal topic, and the study topic is very interesting.  The investigation is well conducted and the present paper is recommended for publication to the present journal after minor revision.

Abstract

The section should include a sentence that reflect the state-of-art background in this field.

Introduction

1.      “A dental implant is a medical device made of alloplastic, biocompatible material that is  inserted into the bone tissue of the jaws to serve as a support for a fixed prosthesis or to  stabilise a removable prosthesis”. The author could describe the more recent insight in dental implant materials and their biocompatibility features (titanium alloys, zirconia implants…).

2.      In addition a concise sentence regarding the implant surface treatment could be effective for the present study topic.

Materials and methods

1.      The inclusion /exclusion criteria should be considered as a separates sub-paragraph.

2.      In this way, the author should  describe the operator’s calibration for the clinical measurements.

3.      “Interventionary studies involving animals or humans, and other studies that require ethical approval, must list the authority that provided approval and the corresponding ethical approval code.” This section should be moved in the statement section part.

Results

A summary table with the implant size and position is suggested in this part.

If applicable, the author could include more clinical photographs of the surgical procedure.

Discussion

The authors should discuss the limitations of the present study.

The null-hypothesis should be discussed in this part of the manuscript.

I suggest to the authors to improve the discussion section after read the follow paper: PMID: 27420108

References

The reference section should be revised according to the journal guidelines.

Author Response

Dear reviewer,

Thank you for your comments and valuable suggestions, according to which we have made some changes in all sections of the paper.

We remain at your disposal for further clarifications.

REVIEWER 1

Abstract 

The section should include a sentence that reflect the state-of-art background in this field. 

We have not added any further considerations (background) within the abstract to avoid straying into the number of words required by the journal at the expense of specifics regarding the results obtained.

Introduction

  1. “A dental implant is a medical device made of alloplastic, biocompatible material that is inserted into the bone tissue of the jaws to serve as a support for a fixed prosthesis or to  stabilise a removable prosthesis”. The author could describe the more recent insight in dental implant materials and their biocompatibility features (titanium alloys, zirconia implants…).
  2. In addition a concise sentence regarding the implant surface treatment could be effective for the present study topic.

According to your suggestions, the introduction was improved with additional explanatory paragraphs supported by citations.

Materials and methods

  1. The inclusion /exclusion criteria should be considered as a separates sub-paragraph.

According to your suggestions, the inclusion and exclusion criteria have been divided into two sub-sections.

  1. In this way, the author should  describe the operator’s calibration for the clinical measurements. 

Specifics concerning calibration and measurement are included in the follow-up section.

  1. “Interventionary studies involving animals or humans, and other studies that require ethical approval, must list the authority that provided approval and the corresponding ethical approval code.” This section should be moved in the statement section part.

The specifications for the ethics committee are included in both sections of the manuscript.

Results

A summary table with the implant size and position is suggested in this part.

In Table 1 we have included the implant placement sites.

The diameter, as specified, was always the same.

The length was not specified because it was not a prerequisite for the purpose of the study objective (marginal bone loss analysis).

If applicable, the author could include more clinical photographs of the surgical procedure. 

Clinical photographs have been included in other studies we have published; in this one we have focused on the analytical aspect of comparing connections.

Discussion 

The null-hypothesis should be discussed in this part of the manuscript. 

I suggest to the authors to improve the discussion section after read the follow paper: PMID: 27420108

The null hypothesis was added both in the statistical analysis section and in the discussion by comparing the data obtained with those of pre-existing studies on the topic. 

References

The reference section should be revised according to the journal guidelines.

The citations were revised according to the journal's guidelines.

Author Response

Dear reviewer,

Thank you for your comments and valuable suggestions, according to which we have made some changes in all sections of the paper.

We remain at your disposal for further clarifications.

Reviewer 3 Report

There are too many inaccuracies in the content.

Starting from the very nature of the study, where on the one hand the authors point to its retrospective nature and then describe it as in the case of a prospective study. Going further, the authors state in the title that the study is 6 years follow-up - material and methods it clearly states "These parameters were measured at implant placement, at 12 months and at 24 months" Here are some more detailed comments:

1.     The introduction is too general, touches very superficially many issues. Needs definite improvement.

2.     References 2, 3, 4 should be replaed with the reference to Branemark original study and a most current deffiniton of osteointegration by Albrektsson T et. al.

3.     Line 49 – 51 „From a structural point of view, the osseointegration process is characterised by three 49 phases: primary stability, secondary stability and maintenance of osseointegration under 50 functional loading”  - A Reference is missing.

4.     Lines 66-69 are very general. I wonder if they are needed.

5.     Line 73: “There are different types of connections: external and internal” there are much more that: Platform switching or platform matching; the type of connection between the abutment and the implant where a Morse taper, tri-channel, external-hex or internal-hex connection can be used

6.     Line 85: Peri-implant marginal bone loss can be influenced by several factors  - How about soft tissue thickness ?

7.      Line 89-90 Reference is needed – statement is very general.

8.     Line 108: Analyzing the study, it seems that it was originally prospective. Were the patients part of a clinical trial, if so, which one and provide a reference. If so, it is necessary to provide details of such examination, including the period when the implants were inserted.

9.     Material and methods: All surgical procedures were performed by the same surgeon with advanced surgical experience - which author, contributions are not clear on this point (please give initial)

10.  Does the 6 years given in the title apply to all patients? The time interval 2016 -2022 in which patients were enrolled would indicate that implants were introduced 2010 - 2016 is this correct ?

11.  Paragramph 2.3-2.5 Such a detailed description of the implant insertion (surgical procedure) in a retrospective study is not needed. As A retrospective study is performed a posteriori - please focus on the activities that were actually performed in the study.

12.  The authors state in the title that the study is 6 years observation ? (prospective or retrospective ? – not clear), however in material and methods it clearly states "These parameters were measured at implant placement, at 12 months and at 24 months" ??!!

13.  numbers don't match... or the results a poorly presented. 66 patients received 69 implants, and further the authors state that in this group 20 solo implants and 46 bridges (i.e. more than 1 unit?)??!! 20 solo implant + 46 bridges (at least 2 units) seems to be a total of 112 implants ?? Similar for the second group.

Author Response

Dear reviewer,

Thank you for your comments and valuable suggestions, according to which we have made some changes in all sections of the paper.

We remain at your disposal for further clarifications.

  1. The introduction is too general, touches very superficially many issues. Needs definite improvement.

According to your suggestions, the introduction was improved with additional explanatory paragraphs supported by citations.

  1. References 2, 3, 4 should be replaed with the reference to Branemark original study and a most current deffiniton of osteointegration by Albrektsson T et. al. 

The references have been replaced.

  1. Line 49 – 51 „From a structural point of view, the osseointegration process is characterised by three 49 phases: primary stability, secondary stability and maintenance of osseointegration under 50 functional loading”  - A Reference is missing.

The references have been added.

  1. Lines 66-69 are very general. I wonder if they are needed.

References have been added to justify the citation. The introduction has been expanded to provide additional information.

  1. Line 73: “There are different types of connections: external and internal” there are much more that: Platform switching or platform matching; the type of connection between the abutment and the implant where a Morse taper, tri-channel, external-hex or internal-hex connection can be used

We have included a more specific sentence including the different types to be suggested, specifying which ones we have focused on in this study.

  1. Line 85: Peri-implant marginal bone loss can be influenced by several factors  - How about soft tissue thickness ? 

We have specified the thickness in the sentence, which was previously implied in the inserted references.

  1. Line 89-90 Reference is needed – statement is very general. 

The references have been added.

  1. Line 108: Analyzing the study, it seems that it was originally prospective. Were the patients part of a clinical trial, if so, which one and provide a reference.If so, it is necessary to provide details of such examination, including the period when the implants were inserted.

As specified in the inclusion criteria, only patients with 6-year follow-up were selected.

  1. Material and methods: All surgical procedures were performed by the same surgeon with advanced surgical experience - which author, contributions are not clear on this point (please give initial)

The surgeries were performed by Professor Paolo Capparè, a maxillofacial surgeon who was also involved in the present study.

  1. Does the 6 years given in the title apply to all patients? The time interval 2016 -2022 in which patients were enrolled would indicate that implants were introduced 2010 - 2016 is this correct ? 

It is correct.

  1. The authors state in the title that the study is 6 years observation ? (prospective or retrospective ? – not clear), however in material and methods it clearly states "These parameters were measured at implant placement, at 12 months and at 24 months"??!!

These parameters were measured at implant placement, at 12 months and once a year during the following period.

Patients are followed up in a standardised protocol and with check-ups at regular periods to ensure monitoring.

  1. numbers don't match...or the results a poorly presented. 66 patients received 69 implants, and further the authors state that in this group 20 solo implants and 46 bridges (i.e. more than 1 unit?)??!! 20 solo implant + 46 bridges (at least 2 units) seems to be a total of 112 implants ?? Similar for the second group.

As shown in the table, 46 is the number of implants inserted to support the bridges and not the number of bridges. The same applies to the second group.

Reviewer 4 Report

The paper has a clear experimental design, a long follow-up period, and comprehensive inclusion and exclusion criteria. Finally, reliable experimental conclusions are drawn, with good quality.

Here are some potential problems:

1. There seems to be no specific description of time variables (T1 and T2) in the paper, and no statistical difference calculation of time variables.

2. In the discussion section, the author did not discuss and analyze the experimental results and other factors that may affect the experimental results, which is lacking in depth, with only similar experimental conclusions in recent years listed.

Author Response

Dear reviewer,

Thank you for your comments and valuable suggestions, according to which we have made some changes in all sections of the paper.

We remain at your disposal for further clarifications.

  1. There seems to be no specific description of time variables (T1 and T2) in the paper, and no statistical difference calculation of time variables.

By comparing the marginal bone loss of group A and group B at the end of the follow-up period, it was possible to assess the impact of the time variable on the marginal bone loss of the groups.

A specification was included in the text to make it clearer.

  1. In the discussion section, the author did not discuss and analyze the experimental results and other factors that may affect the experimental results, which is lacking in depth, with only similar experimental conclusions in recent years listed.

Additional variables have been included in the text; the introduction has been expanded and so have the references.

Round 2

Reviewer 3 Report

It is still important to determine the nature of the study - the authors described the study as retrospective. The description fits a prospective study.

In prospective studies, individuals are followed over time and data about them is collected as their characteristics or circumstances change, and patients are followed-up during the period of time. In retrospective studies, we assess the impact of an intervention that took place in the past.

In the case of the described study, the authors clearly present the inclusion and exclusion criteria and the allocation of patients to groups BEFORE the start of implant treatment - therefore it is prospective?

However, there are no criteria under which they were allocated to the internal/external hex group? was it randomized?

In materials&methods, the authors describe: "The first incision was made on the top of the alveolar crest, shifted on the palatal side to obtain the same level of keratinized mucosa on both flap sides." Was the minimum HKT (Height of Keratinized Tissues?) measured ? Recent soft tissue studies Linkevicius et al. , Hadzik et al. show a tricky methodology and long-term results showing the importance of soft tissue.

Authors should use track-changes to show changes made.

Author Response

Dear reviewer,

thank you for your suggestions. 

I have answered you point by point as follows.
In the text we have highlighted in red the sections expanded and corrected by the other reviewers as well.

We remain at your disposal for further clarification. 

In prospective studies, individuals are followed over time and data about them is collected as their characteristics or circumstances change, and patients are followed-up during the period of time. In retrospective studies, we assess the impact of an intervention that took place in the past.

In the case of the described study, the authors clearly present the inclusion and exclusion criteria and the allocation of patients to groups BEFORE the start of implant treatment - therefore it is prospective?

Dear Reviewer, thank you for your suggestions. The study design is retrospective: we analyzed retrospectively the casistics of our department wih particular inclusion and exclusion criteria, taken from clinical and radiographic recordings. The protocols, for such rehabilitations, are always the same, but the study is not prospective.

However, there are no criteria under which they were allocated to the internal/external hex group? was it randomized?

It is not randomized, because the study is retrospective

In materials&methods, the authors describe: "The first incision was made on the top of the alveolar crest, shifted on the palatal side to obtain the same level of keratinized mucosa on both flap sides." Was the minimum HKT (Height of Keratinized Tissues?) measured ? Recent soft tissue studies Linkevicius et al. , Hadzik et al. show a tricky methodology and long-term results showing the importance of soft tissue. 

Dear Reviewer thank you for your suggestions. Keratinized mucosa was in each case more than 2 mm. We added in the text

Reviewer 4 Report

The manuscript has revised suggestions mentioned last time.

Author Response

Dear auditor, thank you again for your valuable suggestions. 

Round 3

Reviewer 3 Report

The study can be prospective or retrospective. Follow-up study is generally a prospective study because it observes patients undergoing a particular procedure for a certain amount of time ?

Methods are still not clear.

If the same group of patients who are currently being analyzed were enrolled in a clinical trial, allocated within the trial to internal-hex/external-hex groups and then analyzed over a longer period of time, then it is a prospective study and theri follow-up.

If the authors chose among all patients documentation whe were treated in the clinic, who had implants placed  at a specific time, and after picking up random patients based on theri medical history they choose internal-hex/external-hex cases this would be a retrospective study. A retrospective study is performed a posteriori, using information on procedures that were done. 

Author Response

The study can be prospective or retrospective. Follow-up study is generally a prospective study because it observes patients undergoing a particular procedure for a certain amount of time ?

Dear Reviewer thank you for your suggestions. The present study is retrospective. Patients were treated before the beginning of the study, that was approved by the local ethical committee of San Raffaele Hospital as retrospective

Methods are still not clear. 

If the same group of patients who are currently being analyzed were enrolled in a clinical trial, allocated within the trial to internal-hex/external-hex groups and then analyzed over a longer period of time, then it is a prospective study and theri follow-up. 

Patients were included after the treatment: we analyzed our casistics and according to inclusion criteria we selected the patients.

If the authors chose among all patients documentation whe were treated in the clinic, who had implants placed  at a specific time, and after picking up random patients based on theri medical history they choose internal-hex/external-hex cases this would be a retrospective study. A retrospective study is performed a posteriori, using information on procedures that were done. 

Yes, absolutely agree with you. This is what we did, after permission conceived by our ethical committee. We chose among patients treated at our dental clinic, implants placed in a specific period of time, etc. The study isn’t prospective and we analyzed both procedures than radiographical / clinical data that were already acquired.

I hope now everything it’s clear

All my best regards.